# *S. aureus* and *K. pneumoniae* on the Surface and within Core of Tonsils in Adults with Recurrent Tonsillitis

**DOI:** 10.3390/medicina57101002

**Published:** 2021-09-22

**Authors:** Renata Klagisa, Juta Kroica, Ligija Kise

**Affiliations:** 1Department of Otorhinolaryngology, Riga Stradins University, LV-1007 Riga, Latvia; 2Department of Otorhinolaryngology, Daugavpils Regional Hospital, LV-5401 Daugavpils, Latvia; 3Department of Biology and Microbiology, Riga Stradins University, LV-1007 Riga, Latvia; juta.kroica@rsu.lv; 4Department of Doctoral Studies, Riga Stradins University, LV-1007 Riga, Latvia; profesorekise@gmail.com

**Keywords:** *Klebsiella pneumoniae*, *Staphylococcus aureus*, surface and core, tonsillitis

## Abstract

*Background and Objectives*: Recurrent tonsillitis is an infection of the palatine tonsils. Samples for microbiological testing are usually obtained from the inflamed surface of the tonsils. Colonizing the surface bacteria does not always correlate with pathogens causing recurrent tonsillitis and there is no consensus or this in research studies. The aim of the study was to compare whether *Staphylococcus aureus* (*S. aureus*) and *Klebsiella pneumoniae* (*K. pneumoniae*) differ when isolated from the tonsillar surface or tonsillar crypts in patients with recurrent tonsillitis. *Materials and Methods*: a case series study was conducted at a tertiary referral center among 25 patients diagnosed with recurrent tonsillitis. An evaluation of *S. aureus* and *K. pneumoniae* incidence, biofilm formation and antibacterial susceptibility was performed. *Results*: There was a statistically significant association between surface and punch biopsy samples for *S. aureus* (Fisher’s Exact test *p* = 0.004) and *K. pneumoniae* (Fisher’s Exact test *p* < 0.001). A McNemar test did not reveal a statistically significant association. Although the antibacterial resistance profile was not broad, five out of nine *S. aureus* isolates were biofilm producers and four out of five *K. pneumoniae* isolates were biofilm producers. *Conclusions*: Surface and core cultures of tonsils are comparable with a differing incidence between the surface and the punch biopsy cultures for *S. aureus* and *K. pneumoniae.* A larger quantity of bacteria exist in surface samples suggesting that a biopsy sample may be less challenging in evaluating recurrent tonsillitis. We recommend that antibacterial susceptibility results are considered alongside the biofilm-forming potential of isolated bacteria.

## 1. Introduction

Recurrent tonsillitis is an infection of the palatine tonsils, which is characterized by recurrent episodes of tonsillitis, resulting in continuous discomfort in the throat, collections of debris in the palatine crypts, halitosis and cervical lymphadenopathy. Acute symptoms (episodes of tonsillitis) are induced by bacteria released from tonsillar crypts [1] Chronic symptoms are caused by bacteria sheltered within the tonsillar crypts leading to prolonged interaction with the host’s immune system [1]. 

Samples for microbiological testing are usually obtained from the inflamed surface of the tonsils by rotating sterile cotton wool swabs over the surface, avoiding any other part of the oropharynx [2]. Only superficial microorganisms that are highly variable are analyzed and contamination with the normal oral microbiota cannot be ruled out [3]. Colonizing surface bacteria do not always correlate with pathogens causing recurrent tonsillitis and there is no consensus for this in research studies [2,4,5]. Patients’ microbiota can be affected by recent hospitalization or travelling, concomitant diseases or the use of antibiotics. The living environment of patients is important due to variable microbiological changes associated with urban and rural living environments. The study by Khadilkar and Ankle (2016) showed that anaerobic organisms known to inhabit the surface were also present at the core of tonsils and the same antibiotics were efficient against both the surface and core bacteria [5]. The time frame for the acquisition of clinical specimens is limited due to pathogenic microorganisms being present on the surface of tonsils only during exacerbations of tonsillitis. It is not possible to obtain the contents of crypts with a cotton swab. Crypts are narrow passages that penetrate the tonsils. There are microbes in the depth of the crypts. It is recommended to take the clinical specimen from the upper pole of the palatine tonsil, where the largest crypt (*crypta magna* in Latin) is located and where the most common complication, peritonsillar abscess, occurs [6]. Tonsillar infection may stem from bacteria within tonsillar crypts or parenchyma rather than from those on the surface [5].

The objective of this study is to compare whether *Staphylococcus aureus* (*S. aureus*) and *Klebsiella pneumoniae* (*K. pneumoniae*) differ when isolated from the tonsillar surface or tonsillar crypts in patients with recurrent tonsillitis. 

## 2. Materials and Methods

Case series study was performed in Pauls Stradins Clinical University Hospital (PSCUH) in Riga, Latvia, from August 2020 to September 2020. The study protocol was approved by the Ethics Committee of Riga Stradins University (document No. 49/30.11.2017.). Written informed consent was obtained from all subjects before the study. Those with recent antibacterial treatment or those who had failed to give consent were excluded. There was no control group as the samples were taken from patients with a known history of recurrent tonsilitis that were referred to the Otorhinolaryngology Department of PSCUH for scheduled tonsillectomy. A brush sample from uninflamed tonsillar surface was taken prior to surgery and a punch biopsy of tonsillar crypts was performed during surgery in 25 adults undergoing tonsillectomy for recurrent tonsillitis. 

For research purposes, the punch biopsy needle was designed with a prolonged and curved handle and a circular blade for tonsillar crypt biopsy (patent number: LVP2020000055) [7]. 

The brush samples and punch biopsy samples were inoculated on mannitol salt agar and MacConkey agar for *S. aureus* un *K. pneumoniae* to be isolated. The isolated bacteria were identified using VITEK-2 Compact (bioMériux, France). 

Microtiter-plate method was used for in vitro cultivation and quantification of bacterial biofilms [8]. The optical density of the layer of adherent biofilm formed in the microtiter-plate was measured using a microtiter-plate reader (Tecan Infinite F50, Mannedorf, Switzerland, with Magellan™ reader control and data analysis software V 6.6). 

Antibacterial susceptibility tests were performed, and the results were evaluated in accordance with the recommendations of the European Committee on Antimicrobial Susceptibility Testing (EUCAST) ‘Clinical breakpoints and dosing of antibiotics’ (Version 10.0, January 2020) [9]. Overnight cultures were suspended in physiological saline to 0.5 McFarland units (McFarland Densitometer DEN-1, Biosan, Latvia). The suspension was inoculated on Mueller-Hinton agar (Oxid, UK). Selected antibiotics were placed on the inoculated plates and included ceftazidime 10 µg, ampicillin 10 µg, cefotaxime 5 µg, meropenem 10 µg, imipenem 10 µg, amikacin 30 µg, ciprofloxacin 5 µg, chloramphenicol 30 µg, amoxicillin + clavulanic acid 30 µg, and piperacillin + tazobactam 36 µg (Liofilchem, Italy).

## 3. Results

Twenty-five patients were included in this study that comprised of four male and 21 female patients. They were aged between 20 and 71 years, with a median age of 31 at the time of data collection. Seventeen patients were from Riga, three patients from Jelgava, and five patients from other cities (Engure, Bauska, Salaspils, Limbazi and Tukums) in Latvia.

Sixteen patients had concomitant diseases: one patient had bronchial asthma; one patient had a renal abscess and nephrectomy in their medical history; one patient had psoriasis; one patient had pheochromocytoma; three patients had primary arterial hypertension (PAH); one patient had migraine and cystitis in their medical history; one patient had gastritis; one patient had gastroesophageal reflux disease (GERD); one patient had GERD and dysregulation of glucose metabolism; one patient had PAH, Graves’ disease and myocarditis in their medical history; one patient had PAH and uterine fibroids; one patient had GERD, Lyme disease and gout; one patient had tick-borne encephalitis in their medical history; one patient had polyarthritis and hepatitis; and nine patients denied any concomitant diseases.

Twenty-one patients had had recurrent episodes of tonsillitis during the past 3 years, four patients had recurrent episodes of tonsillitis during childhood and had chronic discomfort in the throat, tonsilloliths and halitosis, as adults. Two patients had peritonsillar abscess and one patient had peritonsillitis in their medical history. Four patients had had cryptolysis with radiofrequency, three patients had had cryotherapy before surgery. The most recent antibacterial treatment was received 1 week and 2 years before surgery, with a median time of 6 months. Four patients were abroad 3 weeks to 2 months before the scheduled tonsillectomy. Ten patients were hospitalized (one in January 2020, two in 2018, four in 2017, one in 2016, one in 2013, one in 2008), one patient worked in the hospital as medical staff member, and one patient worked at a cattle breeding farm.

C-reactive protein was tested in 18 patients and was between 0.1 and 6 mg/L with a median of 0.82 mg/L. The white blood cell count was tested in 25 patients and was between 3080 and 9150, with a median of 5760. Five patients had anti-streptolysin O tested, which obtained the measurements 68.1, 277, 96.29, 44.95 and 90 Iu/mL. The rheumatoid factor was tested in six patients, which obtained the measurements 19, 7.8, 11.06, 19, 6.9, and 3. The erythrocyte sedimentation rate was tested in 14 patients, and was measured as 2 and 25 with a median of 3.

*S. aureus* was isolated from brush samples of 12 patients and from the punch biopsy specimens of nine patients (Table 1). Fisher’s Exact test revealed a statistically significant association between *S. aureus* isolation from brush and punch biopsy samples (*p* = 0.004). *K. pneumoniae* was isolated from the brush samples of four patients from the punch biopsy specimens of five patients (Table 1). Fisher’s Exact test revealed a statistically significant association between *K. pneumoniae* isolation from brush and punch biopsy samples (*p* < 0.001). A McNemar test did not reveal a statistically significant association.

*S. aureus* isolates detected in punch biopsy were tested for their biofilm-producing ability (Table 2). Five out of nine *S. aureus* isolates were biofilm producers, two isolates were strong biofilm producers, three isolates were weak biofilm producers, but four out of nine *S. aureus* isolates did not produce a biofilm. Four out of five *K. pneumoniae* isolates were weak biofilm producers and one out of five *K. pneumoniae* isolates were biofilm non-producers.

*S. aureus* and *K. pneumoniae* isolates detected in punch biopsy samples were tested for antibacterial susceptibility. Seven out of nine *S. aureus* isolates were resistant to benzylpenicillin and ampicillin; five out of seven were biofilm producers. All *S. aureus* isolates were intermediate to ciprofloxacin, and sensitive to ampicillin-sulbactam, amoxicillin-clavulanic acid, amikacin, erythromycin, clindamycin, and chloramphenicol. All *K. pneumoniae* isolates were resistant to benzylpenicillin, ampicillin and erythromycin, but sensitive to ampicillin-sulbactam, amoxicillin-clavulanic acid, piperacillin-tazobactam, ceftazidime, cefotaxime, ceftriaxone, meropenem, imipenem, amikacin and ciprofloxacin.

There were no significant associations between recurrent episodes of tonsillitis and concomitant diseases (Fisher’s exact test, *p* = 0.542); recurrent episodes of tonsillitis and the presence of *S. aureus* in punch biopsy samples (Fisher’s exact test, *p* = 0.260), or the presence of *K. pneumoniae* in punch biopsy samples (Fisher’s exact test, *p* > 0.999); recurrent episodes of tonsillitis and biofilm production of *S. aureus* (Fisher’s exact test, *p* = 0.238), or biofilm production of *K. pneumoniae* (Fisher’s exact test, *p* = 0.617); and recurrent episodes of tonsillitis and resistance of *S. aureus* (Fisher’s exact test, *p* = 0.294), or the resistance of *K. pneumoniae* (Fisher’s exact test, *p* > 0.999) (Table 3.).

Tonsillectomy specimens were sent for routine histopathological examination. The histopathological evaluation reports were reviewed and the diagnosis was chronic nonspecific tonsillitis in all patients.

## 4. Discussion

The median age of patients in this study was 31; they were older compared to previous studies, in which patients were predominantly adolescents aged 11 to 20 (44%), followed by children (41%) [5], with the median age being 24 [3]. This study was carried out in Pauls Stradins Clinical University Hospital where only adult patients (≥18 years old) were admitted, therefore explaining the differences in patients’ age. Our study, like other studies before [3,5] had predominantly female participants. In a study by Khadilkar and Ankle, which included 100 patients of chronic tonsillitis, female predominance was explained with an increased health awareness in women [5].

The microbiota of patients and the *S. aureus* carriage can be affected by factors such as working on a farm, recent hospitalization or travelling, concomitant diseases and use of antibiotics. At the time of sampling, patients did not receive antibacterial treatment.

The histories of patients’ concomitant diseases were taken and cross referenced with medical records prior to surgery. At the time of surgery, no exacerbations of chronical illnesses were noted. Some patients’ medical histories showed infectious diseases such as renal abscesses, tick-borne encephalitis, Lyme disease, cystitis, hepatitis, and myocarditis in the past. Inflammatory markers (white blood cell count, C-reactive protein, and erythrocyte sedimentation rate) were within normal range. The specific onset of concomitant diseases in relation to recurrent tonsillitis was not clear due to the lack of specific medical records of such nature and patients’ failure to recall their medical history.

An incidence of surface and core isolates was observed, and an accuracy of culture findings was compared in many studies [2,3,5,10,11]. The accurate identification of the bacterial organism responsible for an infected tonsil might improve culture-directed antibiotic therapy and obviate the need for elective tonsillectomy [11]. In our study we analyzed *S. aureus* as studies proved that *S. aureus* persisted within mucosal biofilms, even intracellularly [12,13,14,15]. *K. pneumoniae* is well known for its biofilm-forming potential. *K. pneumoniae* biofilms can lead to colonization in the respiratory tract, nasopharynx, tonsils [16,17]. In a study by Sarkar et al. (2017), *S. aureus*, group A beta-hemolytic streptococci, and *Klebsiella spp.* were the most common isolates from both surface and core samples with a higher incidence in core samples [11]. Sarkar et al. (2017) concluded that the routine culture of surface swab specimens in patients with recurrent tonsillitis was neither reliable nor valid and recommended core sampling using fine needle aspiration as the diagnostic method of choice [11]. In the current study, the growth of *S. aureus* was more common in brush samples, whereas *K. pneumoniae* was isolated more frequently from punch biopsy samples.

For punch biopsy isolates we provided biofilm growth testing and antimicrobial susceptibility testing. Although the antibacterial resistance profile was not broad, the biofilm growth testing revealed that five out of nine *S. aureus* isolates and four out of five *K. pneumoniae* isolates were biofilm producers. Microorganisms in biofilms were distinctively more resistant to antimicrobial agents and environmental insults and were therefore more difficult to eradicate [1].

## 5. Strengths and Limitations

The punch biopsy needle was developed specifically to obtain the core samples of the tonsils. The report provides the results of the microbiological testing of *S. aureus* and *K. pneumoniae* on the surface and within the core of tonsils in adults with recurrent tonsillitis. Microbiological testing provides information of biofilm forming ability of identified bacteria.

However, several limitations should be addressed. Firstly, the small number of cases observed during the study period could cause bias. Due to the relatively small sample size, we focused mainly on the presence of *S. aureus* and *K. pneumoniae* in tonsillar samples obtained in different ways. Further studies, including those with a larger study group, a control group, an increased bacterial spectrum with biofilm formation and antibacterial susceptibility tests would be necessary to draw more reliable conclusions in terms of tonsillitis. Additionally, a histopathological study of punch biopsy samples would be useful to measure the presence of inflammatory cells and perform a cell count of them. Then, these variables could be compared with the presence of bacteria.

## 6. Conclusions

There was a statistically significant association between the surface and punch biopsy samples for *S. aureus* and *K. pneumoniae*. The surface and core cultures of tonsils were comparable with a differing incidence between the surface and punch biopsy cultures for *S. aureus* and *K. pneumoniae*. A larger quantity of bacteria existed in surface samples suggesting that a biopsy sample may be less challenging in evaluating recurrent tonsillitis. We recommend that antibacterial susceptibility results are considered alongside the biofilm forming potentials of isolated bacteria.

## 7. Patents

Klagisa, R.; Kroica, J.; Kise, L. Punch Biopsy Needle. Patent No: LVP2020000055. In *Izgudrojumi, Preču Zīmes un Dizain-paraugi*. Patent Office of the Republic of Latvia: Riga, Latvia, 2021; Volume 5, pp. 315.

## Figures and Tables

**Table 1 medicina-57-01002-t001:** Growth of *S. aureus* and *K. pneumoniae* from brush and punch biopsy samples.

	Brush Only	Punch Only	Both	Brush Total	Punch Total	Test, *p* Value	
*S.aureus*	4	1	8	12 (25)	9 (25)	Fisher`s Exact test, *p* = 0.004	McNemar test, *p* = 0.375
*K.pneumoniae*	0	1	4	4 (25)	5 (25)	Fisher`s Exact test, *p* < 0.001	McNemar test, *p* > 0.999

**Table 2 medicina-57-01002-t002:** Biofilm production and antibacterial susceptibility profile of *S. aureus* and *K. pneumoniae* isolates obtained with punch biopsy from tonsils of patients undergoing tonsillectomy.

	Biofilm Production		Antibacterial Susceptibility			
	Biofilm Producers	Biofilm Non Producers	Benzylpenicillin	Ampicillin	Ciprofloxacin	Erythromycin
*S.aureus*	5 (9)	4 (9)	R 7 (9)	R 7 (9)	I 9 (9)	0 (9)
*K.pneumoniae*	4 (5)	1(5)	R 5 (5)	R 5 (5)	0 (5)	R 5 (5)

**Table 3 medicina-57-01002-t003:** Characteristics of the study population. Note: PAH, primary arterial hypertension; GERD, gastroesophageal reflux disease; DGM, dysregulation of glucose metabolism; RT, recurrent episodes of tonsillitis during the past 3 years; SA, *S. aureus*; KP, *K. pneumoniae*; 0, biofilm nonproducer; 1, weak producer; 2, moderate producer; 3, strong producer; R, resistance; BP, benzylpenicillin; AMP, ampicillin; E, erythromycin.

N	Age	Sex	Place	Concomitant Diseases	RT	*SA* Brush	*SA* Punch	*KP* Brush	*KP* Punch	*SA* Biofilm	*KP* Biofilm	*SA* Resistance (BP, AMP)	*KP* Resistance (BP, AMP, E)
1	31	♀	Riga	Bronchial asthma	+	+	+	-	-	3		R	
2	27	♀	Engure	Renal abscess, nephrectomy	+	+	+	-	-	1		R	
3	30	♀	Bauska	Psoriasis	+	-	-	+	+		1		R
4	30	♀	Riga	Pheochromocytoma	+	-	+	-	-	3		R	
5	20	♀	Jelgava		+	-	-	-	-				
6	64	♀	Jelgava	PAH	-	+	-	-	-				
7	31	♀	Riga		-	-	-	-	-				
8	26	♀	Riga		+	+	+	-	-	1		R	
9	32	♀	Riga	Migraine, cystitis	+	-	-	-	-				
10	36	♀	Riga		+	+	+	-	-	0		R	
11	31	♀	Riga		+	-	-	-	-				
12	26	♀	Salaspils		+	+	+	-	-	1		R	
13	31	♂	Riga	Gastritis	+	-	-	-	+		0		R
14	25	♀	Riga	GERD, DGM	+	-	-	+	+		1		R
15	71	♀	Riga	PAH, Graves’ disease, myocarditis	+	+	-	-	-				
16	61	♀	Jelgava	PAH, uterine fibroids	+	-	-	-	-				
17	55	♂	Riga	GERD, Lyme disease, gout	-	-	-	+	+		1		
18	34	♀	Limbazi		+	-	-	-	-				
19	22	♀	Riga		+	+	+	-	-	0			
20	55	♀	Riga	PAH	-	-	-	-	-				
21	40	♀	Riga		+	-	-	-	-				
22	38	♂	Tukums	Tick-borne encephalitis	+	+	+	+	+	0	1		
23	27	♂	Riga	PAH	+	+	+	-	-	0			
24	41	♀	Riga	Polyarthritis, hepatitis	+	+	-	-	-				
25	58	♀	Riga	GERD	+	+	-	-	-				

## Data Availability

The data presented in this study are available on request from the corresponding author, through the institutional review board. The data are not publicly available due to restrictions of the institution.

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
