# Peer review of "S. aureus* and *K. pneumoniae* on the Surface and within Core of Tonsils in Adults with Recurrent Tonsillitis"

_medicina, 2021, doi:10.3390/medicina57101002_

Round 1

Reviewer 1 Report

I have found some mistakes:

Literature

  1. Clinical breakpoints and dosing of antibiotics. 2020. – very difficult to find, not up to date (2021 is actual)
  2. Zautner AE instead of Andreas, E.Z.

The article should be lectured and corrected by an English speaker.

Overall, the article is novel regarding the higher age of the patients. It is a good continuation of the articles cited in the literature.

Author Response

Thank you for valuable and constructive criticism. We addressed Your comments and suggestions point by point. Changes have been marked by green color.

Literature

  1. Clinical breakpoints and dosing of antibiotics. 2020. – very difficult to find, not up to date (2021 is actual)

Response: Thank you, the additional information (link for EUCAST tables, pages for actual bacteria) is added to the Literature. Since I did the research in 2020, I used the 2020 clinical breakpoints and indicated the version used.

  1. Anonymous. The European Committee on Antimicrobial Susceptibility Testing. Breakpoint tables for interpretation of MICs and zone diameters. Version 10.0, 2020, 34 – 38, 84 - 87.

Available from: https://www.eucast.org/ast_of_bacteria/previous_versions_of_documents/

  1. Zautner AE instead of Andreas, E.Z.

Response: Thank you, this is corrected now.

  1. Zautner, A.E. Adenotonsillar Disease. Recent Pat Inflamm Allergy Drug Discov. 2012, 6, 121–9.
  2. Zautner, A.E.; Krause, M.; Stropahl, G.; Holtfreter, S.; Frickmann, H.; Maletzki, C.; Kreikemeyer, B.; Pau, H.W.; Podbielski, A.; Bereswill, S. Intracellular Persisting Staphylococcus aureus Is the Major Pathogen in Recurrent Tonsillitis. Bereswill S, editor. PLoS ONE. 2010, 5, e9452.

Reviewer 2 Report

This manuscript is related with the association between S.aureus and K. pneumoniae in recurrent tonsillitis, in which the presence of these bacteria in tonsils and recurrent tonsillitis is compared. 

Overall, this manuscript is interesting due to the comparison of both bacteria in tonsils disease.

Comments

1. Lines 56-58: The objective could be redacted in a better manner, due to it seems a description of methodology than to actual objective.

Material and methods

2. Line 62, Was only evaluated one month in this study? Why did this happen?

3. Line 65, It is not entirely clear whether the authors included a control group or not, but in their objective, they mentioned a study group of tonsils, crypts and tonsillitis. Thus, are the tonsils and tonsillar crypts control groups? Please review this.

Were the samples of punch biopsy inoculated as brush samples? Why were two types of samples, what is the differences? Please explain with detail in the methodology. 

Why did not they do a histopathological study of the punch sample in order to measure the presence of inflammatory cells and perform a cell count of them?   Results 

Line 90. "A case series study was conducted on 25 patients with recurrent tonsillitis" this line was repeated, please review it.

Line 92. "17 patients" What was the importance of mentioning the origin of residence of the patients? Please explain this in the introduction.

Line 94. "Engure" This mention is interesting, will it be possible to explain the activities or working place of the rest of the patients? I consider it could be interesting to relate the working activities with recurrent tonsillitis.   Line 95 "16 patients" Will it be possible to explain these interesting data in a table and to relate their diseases with episodes of tonsillitis and other features described in the paragraph? (Relate concomitant disease with the tonsils disease and the variables describes on results) Then, to compare these variables with the presence of bacteria. (S.aureus, K. pneumoniae) and, if possible, to relate the variables described  (disease and association with tonsils diseases) with the antibacterial susceptibility.

Discussion

Lines 148 to 150: I did not understand these lines, authors said that predominance of tonsillitis were related to increase of awareness. Please explain it better. 

Lines 151 to 154: "Microbiota" Authors said that S. aureus is related to working on a farm, why was this claimed if only one patient was working on a farm?

Line 155 to 160: "Concomitant disease" I did not understand this paragraph.

Did the patients acquired these diseases when they developed tonsillitis? Please explain it better.

Conclusions

The conclusion must include the advantages, limitants and contributions that authors make with this manuscript to current knowledge, please review and remake it.

Overall comments This manuscript is interesting since the authors studied the presence of two important bacteria related to tonsils diseases, specifically on recurrent tonsillitis. Although, this study is interesting, the authors need to review  the entire paper and perform a better relationship between variables. This manuscript seems to be focused on  describing patients with tonsils disease and does not relate adequately the included variables. Thus, I recommend to deepen into the relationship of concomitant disease, bacteria, and recurrent tonsillitis to improve this interesting manuscript.  Furthermore,  I recommend to make tables related to clinical variables and figures.

Author Response

Thank you for valuable and constructive criticism. We addressed Your comments and suggestions point by point. Changes have been marked by green color.

  1. Lines 56-58: The objective could be redacted in a better manner, due to it seems a description of methodology than to actual objective.

Response: Thank you! The objective was updated.

Objective is to compare whether Staphylococcus aureus (S. aureus) and Klebsiella pneumoniae (K. pneumoniae) differ when isolated from tonsillar surface or tonsillar crypts in patients with recurrent tonsillitis.

Material and methods

  1. Line 62, Was only evaluated one month in this study? Why did this happen?

Response: Patient`s data and tonsillar samples were obtained and analyzed from August 2020 to September 2020. It was the time necessary for 25 patients to be included in the study. However, the study protocol was designed earlier. The study protocol was approved by the Ethics Committee of Riga Stradins University in November 2017. The patent of punch biopsy needle was approved in May 2021.

  1. Line 65, It is not entirely clear whether the authors included a control group or not, but in their objective, they mentioned a study group of tonsils, crypts and tonsillitis. Thus, are the tonsils and tonsillar crypts control groups? Please review this.

Response: Thank you! The aspect of a control group was reviewed.

There was no control group as the samples were taken from patients with known history of recurrent tonsillitis that were referred to Otorhinolaryngology Department for scheduled tonsillectomy.

Were the samples of punch biopsy inoculated as brush samples? Why were two types of samples, what is the differences? Please explain with detail in the methodology.

Response: Per Your recommendation details in the methodology were added.

The brush samples and punch biopsy samples were inoculated on mannitol salt agar and MacConkey agar for S.aureus un K.pneumoniae to be isolated.

Why did not they do a histopathological study of the punch sample in order to measure the presence of inflammatory cells and perform a cell count of them?   Results

Response: Since focus of our study was on microbiological factors immunohistochemical data are not available. However, per Your recommendations we have added results of routine histopathological evaluation reports. We have also addressed study limitations.

Results. Tonsillectomy specimens were sent for the routine histopathological examination. The histopathological evaluation reports were reviewed. The histopathological diagnosis was chronic nonspecific tonsillitis in all patients.

Strengths and Limitations. Additionally, a histopathological study of punch biopsy samples would be useful to measure the presence of inflammatory cells and perform a cell count of them. Then, to compare these variables with the presence of bacteria.

Line 90. "A case series study was conducted on 25 patients with recurrent tonsillitis" this line was repeated, please review it.

Response: Thank you, this is corrected now.

Twenty-five patients were included in this study that comprised 4 male and 21 female patients.

Line 92. "17 patients" What was the importance of mentioning the origin of residence of the patients? Please explain this in the introduction.

Response: Per Your recommendation explanation were added in the Introduction.

Patient`s microbiota can be affected by recent hospitalization or travelling, concomitant diseases or use of antibiotics. The habitat of patients is of an importance due to variable microbiological changes associated with urban and rural living environment.

Line 94. "Engure" This mention is interesting, will it be possible to explain the activities or working place of the rest of the patients? I consider it could be interesting to relate the working activities with recurrent tonsillitis.   Line 95 "16 patients" Will it be possible to explain these interesting data in a table and to relate their diseases with episodes of tonsillitis and other features described in the paragraph? (Relate concomitant disease with the tonsils disease and the variables describes on results) Then, to compare these variables with the presence of bacteria. (S.aureus, K. pneumoniae) and, if possible, to relate the variables described  (disease and association with tonsils diseases) with the antibacterial susceptibility.

Response: Specific working condition data are not available as our focus was towards noting whether patients lived in a rural or urban environment and possible influence from working with cattle. However, per Your recommendations we have added the table of concomitant diseases, episodes of tonsillitis, presence of bacteria and compared these variables.

There was no significant association between episodes of recurrent tonsillitis and concomitant diseases (Fisher`s exact test, p=0.542), episodes of recurrent tonsillitis and presence of S.aureus in punch biopsy samples (Fisher`s exact test, p=0.260) or presence of K.pneumoniae in punch biopsy samples (Fisher`s exact test, p>0.999), episodes of recurrent tonsillitis and biofilm production of S.aureus (Fisher`s exact test, p=0.238) or biofilm production of K.pneumoniae (Fisher`s exact test, p=0.617), episodes of recurrent tonsillitis and resistance of S.aureus (Fisher`s exact test, p=0.294) or resistance of K.pneumoniae (Fisher`s exact test, p>0.999) (Table 3).

Table 3. Characteristics of the study population. Note: PAH, primary arterial hypertension; GERD, gastroesophageal reflux disease; DGM, dysregulation of glucose metabolism; RT, recurrent episodes of tonsillitis during the past 3 years; SA, S.aureus; KP, K.pneumoniae; 0, biofilm nonproducer; 1, weak producer; 2, moderate producer; 3, strong producer; BP, benzylpenicillin; AMP, ampicillin; E, erythromycin.

Discussion

Lines 148 to 150: I did not understand these lines, authors said that predominance of tonsillitis were related to increase of awareness. Please explain it better.

Response: Thank you, this is explained now, the reference is added.

In a study by Khadilkar and Ankle which included 100 patients of chronic tonsillitis female predominance was explained with increased health awareness in women (5).

Lines 151 to 154: "Microbiota" Authors said that S. aureus is related to working on a farm, why was this claimed if only one patient was working on a farm?

Response: It is important to note possible influencing factors of S.aureus incidence in microbiological sample testing, even if ultimately one patients is subjected to them, same as we mentioned possible link to concomitant diseases, previous hospitalization episodes and more. Since our study group was rather small, we elaborate that in a larger study populous we would see more patients that are influenced by working on farms. 

Line 155 to 160: "Concomitant disease" I did not understand this paragraph.

Did the patients acquired these diseases when they developed tonsillitis? Please explain it better.

Response: Thank you, the paragraph has been rewritten.

A history of patients’ concomitant diseases was taken and cross referenced with medical records prior to surgery. At the time of surgery, no exacerbations of chronical illnesses were noted. Some patients’ medical history showed infectious diseases such as renal abscess, tick-borne encephalitis, Lyme disease, cystitis, hepatitis, and myocarditis in the past. Inflammatory markers (white blood cell count, C-reactive protein, erythrocyte sedimentation rate) were within normal range. Specific onset of concomitant diseases in relation to recurrent tonsillitis was not clear due to lack of specific medical records of such nature and patients’ failure to recall their medical history.

Conclusions

The conclusion must include the advantages, limitants and contributions that authors make with this manuscript to current knowledge, please review and remake it.

Response: Thank you. Strengths and limitations of this study have been addressed in a separate paragraph.

  1. Strengths and Limitations

            Punch biopsy needle was developed specifically for obtaining core samples of tonsils. The report provides the results of microbiological testing of S.aureus and K.pneumoniae on surface and within core of tonsils in adults with recurrent tonsillitis. Microbiological testing provided information of biofilm forming ability of identified bacteria.

However, several limitations should be addressed. Firstly, the small number of cases observed during the study period could cause bias. Due to the relatively small sample size, we have focused mainly on the presence of S.aureus and K.pneumoniae in tonsillar samples obtained in different ways. Further studies, including those with a larger study group, a control group, increased bacterial spectrum with biofilm formation and antibacterial susceptibility tests would be necessary to make more reliable conclusions in terms of tonsillitis. Additionally, a histopathological study of punch biopsy samples would be useful to measure the presence of inflammatory cells and perform a cell count of them. Then, to compare these variables with the presence of bacteria.

Round 2

Reviewer 2 Report

The manuscript has improved considerably in comparison to the prior version. Now is more understandable and better explained.